

# Antimicrobial activity of *Ruta angustifolia* L. Pers against periodontal pathogen: *Porphyromonas gingivalis*

Husna Hazirah Bakri[1], Syarifah Nur Syed Abdul Rahman[1],
Zarith Safinaz Dol Bakri[1], Elly Munadziroh[2] and
Wan Himratul Aznita Wan Harun[1,2]

[1] Department of Oral & Craniofacial Sciences, Faculty of Dentistry, Universiti Malaya,
Kuala Lumpur, Malaysia
[2] Department of Dental Material, Faculty of Dental Medicine, Airlangga University, Surabaya,
East Java, Indonesia

Corresponding authors
Wan Himratul Aznita Wan Harun,
aznita@um.edu.my
Syarifah Nur Syed Abdul Rahman,
synur@um.edu.my

## ABSTRACT

**Background:** *Porphyromonas gingivalis* is widely recognised as a periodontal pathogen. In recent years, there has been growing interest in the use of medicinal plant extracts as alternative treatments for periodontitis to combat the emergence of antibiotic-resistant bacteria. *Ruta angustifolia* L. Pers has been traditionally used to treat various ailments, including oral bacterial infections. However, the antimicrobial potential of *R. angustifolia* extracts against the periodontal pathogen *P. gingivalis* remains unexplored. Hence, the aim of this study was to investigate the antimicrobial activity of *R. angustifolia* extracts against *P. gingivalis*.
**Methods:** The antimicrobial activity of *R. angustifolia* extracts (crude methanol, hexane and chloroform fractionated extracts) against *P. gingivalis* was evaluated using the well diffusion method. Additionally, the minimum inhibitory concentration (MIC) and minimum bactericidal concentration (MBC) were determined. Biofilm biomass assessment and live/dead cell viability assays were performed to analyse the effect of *R. angustifolia* extracts. Ultrastructural morphological changes in *P. gingivalis* cells were determined using field emission scanning electron microscopy (FE-SEM).
**Results:** It was found that *P. gingivalis* was susceptible to *R. angustifolia* extracts, with the chloroform fractionated extract exhibiting the highest inhibition zones. The MIC and MBC of chloroform fractionated extract were determined to be 6.25 mg/mL which substantially reduced *P. gingivalis* biofilm biomass. Live/dead cell viability assays showed the highest percentage of dead *P. gingivalis* cells after 48 h of incubation. FE-SEM confirmed that the chloroform fractionated extract effectively damaged the bacterial cell wall and altered the ultrastructural morphology of *P. gingivalis*.
**Conclusion:** The results indicated that extracts of *R. angustifolia* has the potential to be used as an alternative treatment in addition to conventional periodontal therapies.

## INTRODUCTION

According to the World Health Organisation (WHO), nearly 3.5 billion people worldwide are affected by oral diseases, with three out of four affected individuals living in middle-income countries. Globally, severe periodontal disease is estimated to affect around 19% of the global adult population, representing more than 1 billion cases (*Santhosh et al., 2023*). The global oral care market has witnessed a significant rise, primarily due to the increasing prevalence of oral health issues such as periodontal diseases, dental caries, plaque-induced gingivitis, and oral cancer (*Amil et al., 2024*). Periodontal diseases are intricate, multifactorial conditions characterised by the progressive resorption of the structures that support the teeth. This process involves a variety of microorganisms that trigger inflammation, resulting in damage to the supportive tissues. Consequently, this condition leads to the formation of periodontal pockets, loss of alveolar bone, and ultimately, tooth loss (*Lorenzi et al., 2024*). It is also associated with several systemic health conditions, such as diabetes mellitus and atherosclerosis (*Sanz et al., 2020*). However, treating periodontal disease presents several challenges, including difficulties in early diagnosis, the complexity of managing its multifactorial nature, and variability in patient responses to treatment. Additionally, the bacterial biofilm that forms in periodontal pockets can be highly resistant to conventional therapies, often requiring a combination of mechanical, antimicrobial, and sometimes surgical interventions. Furthermore, patient compliance in maintaining long-term oral hygiene practices plays a critical role in the success of periodontal disease management (*Chang, Kantrong & Darveau, 2021*).

A model of periodontitis pathogenesis has been proposed that emphasises polymicrobial synergy and dysbiosis. Dysbiosis refers to an imbalance within the oral microbial community. In a healthy state, the microbiome is diverse and balanced; however, dysbiosis occurs when harmful bacteria outnumber beneficial ones, leading to inflammation and tissue damage. In the context of periodontal disease, dysbiosis is characterised by an overgrowth of pathogenic bacteria, disrupting microbial homeostasis and promoting chronic inflammation, which ultimately contributes to the destruction of periodontal tissues. According to this model, the disease originates from an imbalance between the host and the oral microbiota, where specific species, such as *P. gingivalis, Tannerella forsythia*, and *Treponema denticola*, collaborate to establish an inflammatory microbiota, resulting in periodontal inflammation (*Belibasakis et al., 2023*).

*P. gingivalis* is considered a predominant pathogenic bacterium associated with periodontitis due to its capacity to disrupt the balance of the oral microbiota and impair host immune responses. This disruption significantly contributes to the pathogenesis of periodontal disease by driving gum inflammation, breaking down periodontal tissues, causing loss of alveolar bone, and forming periodontal pockets, all of which complicate treatment and accelerate the chronic progression of the disease (*Howard, Gonzalez & Garneau-Tsodikova, 2021*). The bacterium possesses various virulence factors, including enzymes that degrade host tissues and molecules that modulate immune responses, enabling it to efficiently colonise the subgingival area and evade immune detection. This not only triggers periodontal disease but also promotes oral dysbiosis, a shift in the

microbial community that exacerbates disease progression. Its ability to alter the local environment by modulating inflammatory pathways further contributes to the chronic nature of the condition. Indeed, colonisation by *P. gingivalis* disrupts the normal balance of oral microbial communities, leading to inflammation and subsequent bone loss, which complicates effective treatment and management (*Benahmed et al., 2022*).

Factors such as poor oral hygiene, tobacco use, and systemic health conditions can contribute to the development of dysbiosis. Consequently, the growing awareness of oral hygiene has prompted individuals to invest in oral care products for themselves and their families. As a result, the oral care market has experienced significant revenue growth, with an estimated cumulative annual growth rate of 8.1%. Among the diverse range of products available, mouthwashes have generated the highest revenue, closely followed by toothpaste (*Siripipatthanakul & Sixl-Daniell, 2021*). Nonsurgical traditional treatment methods for periodontal diseases, such as scaling and root planing (SRP), are insufficient to fully eliminate oral pathogens or prevent their recurrence. Additionally, the use of systemic antibiotics for treating periodontitis is constrained by the necessity of administering high doses to achieve effective concentrations in gingival fluid. This approach is further complicated by the rapid emergence of antibiotic resistance among bacteria and the potential side effects associated with these medications (*Uddin et al., 2021*). Moreover, prolonged use of chemical ingredients in oral care products, such as chlorhexidine, can lead to tooth and tissue staining, as well as other adverse effects, including calculus accumulation, transient taste disturbances, and impacts on the oral mucosa (*Böhle et al., 2022*). Allergic reactions to chlorhexidine may also provoke type IV hypersensitivity responses, which can manifest as a burning sensation in the mouth, erythema of the gingival tissues, and stomatitis. These issues present significant challenges for both healthcare providers and patients (*Kotsailidi et al., 2020*). Given these challenges, there is increasing interest in exploring natural alternatives, particularly herbs and spices known for their antibacterial and chemopreventive properties (*Refaey et al., 2024*). Thus, finding a safe and effective therapeutic agent for oral health is deemed necessary.

Historically, plants and their extracts have been utilised for therapeutic purposes across various cultures, with traditional medicine practices highlighting their benefits. The appeal of herbal extracts lies in their perceived safety, cost-effectiveness, and efficacy, as well as their role as alternatives or complements to conventional treatments (*Abdalla et al., 2022*). Herbal extracts have been employed for centuries to address a variety of health conditions, including oral health issues such as gingivitis, periodontitis, and dental caries. The recent popularity of herbal medicines and products is evident among dental patients, professionals, and a wider range of consumers and healthcare practitioners, due to their potential benefits for oral health and alveolar bone protection (*Mosaddad, Hussain & Tebyaniyan, 2023*). A previous study has indicated that the systemic administration of curcumin, a compound derived from turmeric, significantly reduced alveolar bone loss in a ligature-induced mice model, suggesting its potential as an alternative treatment for initial periodontal therapy (*Wang et al., 2023*). Another study revealed that extracts from *Cinnamomum zeylanicum* and *Salvadora persica* exhibited antibacterial activity against various periodontal pathobionts such as *P. gingivalis*, *T. denticola*, *T. forsythia*, and

*Actinobacillus actinomycetemcomitans* (*Saquib et al., 2019*). The Rutaceae family, particularly the genus Ruta, such as *Ruta angustifolia* L. Pers, commonly referred to as "garuda". This plant has been traditionally used in Indonesia for treating jaundice and liver disorders. Furthermore, it is cultivated in Java and Vietnam, where its decoction is commonly employed to relieve cramps, bloating, and fever (*Noer, Abinawanto & Basith, 2018*).

Previous studies reported that various extracts of *R. angustifolia* exhibit a wide range of biological activities, including strong antiviral effects (*Wahyuni et al., 2014*) and significant cytotoxic activity against several cancer types, such as lung cancer (A549), colon cancer (HCT-116 and HT29), and cervical cancer (CaSki) (*Richardson, Aminudin & Abd Malek, 2017*; *Suhaimi, Hong & Abdul Malek, 2017*). Additionally, *R. angustifolia* has demonstrated antibacterial properties, attributed to naturally occurring compounds like flavonoids, alkaloids, and coumarins. These compounds have shown various medicinal effects, particularly against microbial infections. For example, chalepin, isolated from the leaves of *R. angustifolia*, demonstrated modest antiparasitic activity against *Trypanosoma cruzi*, the pathogen responsible for Chagas disease (*Bailly, 2023*). Furthermore, essential oil extracted from *R. angustifolia* has shown significant antifungal activity against *Fusarium oxysporum* and *Botrytis cinerea* (*Nahar et al., 2021*). However, there is a lack of comprehensive research on the antibacterial effects of *R. angustifolia* extracts against oral bacteria associated with periodontal diseases. Therefore, this study aims to investigate the antimicrobial efficacy of *R. angustifolia* extracts against *P. gingivalis*, a prominent periodontal pathogen, by assessing biofilm biomass, cell viability, and morphological characteristics. The hypothesis of this study posited that extracts of *R. angustifolia* exhibit significant antimicrobial activity against *P. gingivalis*, with the chloroform fractionated extract demonstrating the highest efficacy in inhibiting bacterial growth, reducing biofilm formation, and inducing cell death, thereby suggesting its potential as a viable alternative treatment for periodontitis.

## MATERIALS AND METHODS

### Source of plant material

The plants of *Ruta angustifolia* L. Pers were purchased from Sg Buloh Flowers & Plants Nursery Wholesale, Sungai Buloh, Selangor, Malaysia. A voucher specimen numbered KLU48128, has been deposited at the Herbarium of the Institute of Biological Sciences, Faculty of Science, Universiti Malaya, Kuala Lumpur, Malaysia.

### Preparation of plant extracts

The plant extraction method was performed based on the protocol described by *Suhaimi, Hong & Abdul Malek (2017)*, with slight modifications. A total of 100 g of powdered leaves of *R. angustifolia* was soaked three times in methanol (MeOH), yielding a greenish crude methanol extract (28.80 g, 28.87%). Hexane was used to extract the crude MeOH (30 g), resulting in a hexane insoluble extract (28.85 g, 96.17%) and a hexane fractionated extract (1.15 g, 3.83%). The hexane insoluble extract underwent further extraction with chloroform, producing an aqueous residue and a chloroform fractionated extract (9.05 g,

30.2%). The resulting crude MeOH and fractionated extracts (hexane and chloroform) were evaporated using a rotary evaporator (BÜCHI, Switzerland) at 40 °C under reduced pressure to remove the solvents completely (*Seidel, 2012*; *Suhaimi, Hong & Abdul Malek, 2017*). Prior to assessing antimicrobial activity, the extracts of *R. angustifolia* were reconstituted in 1% dimethyl sulfoxide (DMSO).

## Bacterial strains and culture conditions

*P. gingivalis* ATCC 33277 was grown from stock culture in BHI-T broth (Tryptone Soya Broth & Brain Heart Infusion broth), supplemented with hemin (5 mg/mL), menadione (5 mg/mL) & cysteine (0.50 g/mL), for 48 to 72 h at 37 °C under anaerobic conditions. The oral bacterial strain was harvested from the broth by centrifugation (10,000×*g*, 10 min, 4 °C) and washed three times with phosphate-buffered saline (PBS) prior to the experiments. The optical density at 600 nm ($OD_{600}$) of the bacterial suspensions was adjusted to 0.1 using a UV-VIS spectrophotometer (equivalent to $10^8$ colony-forming units per millilitre, CFU/mL) for *P. gingivalis* in all experiments. Fresh cultures of *P. gingivalis* were Gram-stained for validation and used in all relevant experiments (*Kobatake et al., 2019*; *Paranagama et al., 2020*).

## Well diffusion assay

Antimicrobial activity was determined using the well diffusion method (*Michaylova et al., 2021*). A volume of 100 μL of a $10^8$ CFU/mL suspension of *P. gingivalis* suspension ($OD_{600}$ adjusted to 0.1) was evenly spread over the surface of the BHI-T agar using a sterile cotton swab (*Zou et al., 2022*).

Using a sterile cork borer, wells measuring 6 mm in diameter were punched in the agar and filled with 30 μL of crude MeOH and fractionated extracts of *R. angustifolia* at concentration of 50 mg/mL & 100 mg/mL, respectively. The plates were incubated under anaerobic conditions for 24 to 72 h at 37 °C, based on optimised growth observations (*Michaylova et al., 2021*). A 0.12% chlorhexidine gluconate (CHX) was used as the positive control, while 1% DMSO served as the negative control. The final concentration of DMSO in the test wells did not exceed 1% (v/v). The inhibition zone diameter was measured using a 0–150 mm METR-ISO Electronic Digital Caliper 6″, manufactured by J.P. Selecta company in Spain. All data presented are the mean ± standard deviation (SD) of triplicates obtained from three independent experiments.

## Determination of the minimum inhibitory concentration (MIC) and minimum bactericidal concentration (MBC)

The two-fold microdilution broth method was employed to obtain the MIC value (*Nordin, Wan Harun & Abdul Razak, 2013*). In this test, eight concentrations were tested, with triplicate measurement taken for each concentration. The concentrations tested were 50, 25, 12.50, 6.25, 3.13, 1.56, 0.78 and 0.39 mg/mL. The bacterial cells of *P. gingivalis* were treated with the chloroform fractionated extract of *R. angustifolia* using two-fold serial dilutions. Briefly, 100 μL of BHI-T broth was dispensed into the wells of a 96-microtiter plate, labelled from Well 1 (W1) to Well 8 (W8). Following this, 50 mg/mL of chloroform

fractionated extract of *R. angustifolia* was added into W1, and two-fold serial dilution were performed for the remaining wells. Hence, the final concentrations of the extract ranged from 50 to 0.39 mg/mL.

A 0.12% CHX was used as the positive control. The chloroform fractionated extract of *R. angustifolia* at different concentrations, without the addition of bacterial culture served as the blank. The negative control consisted of wells containing solely a mixture of BHI-T broth and *P. gingivalis* suspension with 1% DMSO. Then, 20 µL of *P. gingivalis* suspension ($10^8$ CFU/mL) was added to W1 through W8. After incubating overnight under anaerobic conditions at 37 °C, bacterial growth (turbidity) was visually assessed and confirmed using a spectrophotometer at an OD of 600 nm. The methods for conducting the MIC and MBC tests were according to the Clinical and Laboratory Standard Institute guidelines (*Clinical and Laboratory Standards Institute (CLSI), 2020*). The MIC was defined as the lowest concentration of the extract in a well without turbidity (*Madhloom et al., 2022*). All data presented are the mean ± standard deviation (SD) of triplicates obtained from three independent experiments.

For the determination of MBC, 50 µL was taken from the MIC assay wells and spread onto fresh BHI-T agar plates. The plates were incubated under anaerobic conditions at 37 °C for 24 to 72 h, or until visible growth appeared (*Fadare et al., 2022*). The MBC was then defined as the minimum concentration of the chloroform fractionated extract from *R. angustifolia* at which no visible microbial growth was observed on the agar plate. To ensure accuracy and reproducibility, all antimicrobial assessments of the extracts were conducted in triplicate at three separate times.

## Crystal violet biofilm assay

The crystal violet staining method was used to analyse the biomass of biofilms using a broth microdilution method in a 96-well microtiter plate, with some modifications (*He et al., 2020*). One hundred microliters of $10^8$ CFU/mL *P. gingivalis* suspension and 100 µL of two-fold serial dilutions of the chloroform fractionated extract of *R. angustifolia* were added to each well in the microtiter plate. The plates were incubated for 48 h under anaerobic condition at 37 °C. The final concentrations of the extract were 12.50, 6.25, 3.13 and 1.57 mg/mL. The wells containing solely the BHI-T broth and *P. gingivalis* suspension mixture served as the negative control. All data presented are the mean ± standard deviation (SD) of triplicates obtained from three independent experiments.

The supernatant was removed, and the wells were washed three times with PBS. Subsequently, the adhering biofilms were stained with 0.04% (w/v) crystal violet for 15 min after incubation with methanol. For detection, 95% ethanol was added after washing three times with deionised water. The OD values were measured at a wavelength of 550 nm (*Sharma & Saharan, 2016*). The percentages of biofilm inhibition against *P. gingivalis* at different concentrations of the chloroform fractionated extract of *R. angustifolia* were calculated as:

$$\% \text{ biofilm inhibition} = [1 - (A_C/A_0)] \times 100$$

where $A_C$ represents the absorbance of the well with treated *P. gingivalis* (with extract) and $A_0$ the absorbance of the control well, untreated *P. gingivalis* (without extract).

## Live/dead cell viability assay using fluorescence microscopy

The assessment of live/dead cell viability assay were conducted according to the protocol by *Luan et al. (2022)*, with slight modifications. Briefly, fresh cultures of *P. gingivalis* were incubated under anaerobic conditions for 48 h at 37 °C. The bacterial suspension was adjusted to an $OD_{600}$ of 0.1 (equivalent to $10^8$ CFU/mL) by diluting it in supplemented BHI-T broth using a UV-VIS spectrophotometer. *P. gingivalis* biofilm was grown as described by *Zou et al. (2022)* with slight modifications. The *P. gingivalis* biofilm was then incubated under anaerobic conditions with or without the chloroform fractionated extract of *R. angustifolia* at concentrations of 3.13 and 6.25 mg/mL in a 12-well plate, maintaining a volume ratio of 1:1 (v/v) for 24 and 48 h, respectively.

At the end of the incubation period, the cells were harvested and washed three times with PBS. The samples were pelleted by centrifugation at 10,000×$g$ for 10 min, and then resuspended in supplemented BHI-T broth. Subsequently, 3 μL of the LIVE/DEAD® BacLight™ Bacterial Viability Kit (L7012; Thermo Fisher Scientific, Waltham, MA, USA) dye mixture, comprising SYTO9 and PI, was added per mL of bacterial suspension. The stained samples were incubated at room temperature in the dark for 15 min. A 5 μL aliquot of the stained suspension was placed onto a glass microscope slide. The slides of untreated and treated *P. gingivalis* biofilms with the chloroform fractionated extract of *R. angustifolia* were observed immediately and recorded using a fluorescence microscope (Nikon Eclipse Ti-E Inverted Microscope) through FITC (Ex/Em = ~465/515 nm) and TRITC (Ex/Em = ~528/590 nm) channel at 10x magnification.

## Field Emission Scanning Electron Microscopy (FE-SEM) to examine cell morphology

Briefly, *P. gingivalis* cultures were incubated under anaerobic conditions for 48 h at 37 °C and adjusted to an $OD_{600}$ of 0.1 to achieve a cell concentration of $10^8$ CFU/mL. The *P. gingivalis* biofilm was then incubated under anaerobic conditions with or without the chloroform fractionated extract of *R. angustifolia* at a concentration of 6.25 mg/mL in a 12-well plate, maintaining a volume ratio of 1:1 (v/v) for 24 and 48 h, respectively (*Luan et al., 2022*).

After 24-h incubation under anaerobic condition at 37 °C, the bacterial suspensions were fixed in 4% glutaraldehyde for over 4 h (*Arvizu & Murray, 2021*). The samples were then washed twice in buffer for 10 min each, followed by fixation in 1% osmium tetroxide for 1 h. After two 10-min washes in double-distilled water (DDH$_2$O), the samples were subjected to a series of dehydration steps in ascending alcohol concentrations from 30, 50, 70, 80, 90, 95 and 100% (twice), with 15 min each step. The samples were further dehydrated with three different ratios (3:1, 1:1, and 1:3) of ethanol-acetone mixtures for 15 min each step, before being fixed in pure acetone for 20 min (twice). Finally, the samples

were critical point dried (CPD) for 1 to 3 h, mounted on stubs with carbon adhesive, sputter-coated with gold, and viewed using a scanning electron microscope (FE-SEM).

## Statistical analysis

All data are presented as the mean ± standard deviation (SD) of triplicates obtained from three independent experiments. Statistical analyses were conducted using SPSS software (version 27.0; IBM Corp., Armonk, NY, USA). The normality of the data for biofilm biomass inhibition of *P. gingivalis* was assessed using the Shapiro–Wilk test, which indicated that the data followed a normal distribution ($p > 0.05$). Data evaluation between groups was analysed using one-way ANOVA, and *post-hoc* Tukey's HSD test was applied for multiple comparisons. The results of inhibitory activities for *R. angustifolia* extracts against *P. gingivalis* were subjected to the Shapiro-Wilk test to assess normality ($p < 0.05$), which indicated that the data distribution was not normal. Statistical analysis between groups was performed by applying the non-parametric Kruskal-Wallis test, followed by Dunn's *post-hoc* test. The significance level was set at 0.05 for all analyses, with *p*-values < 0.05 considered statistically significant.

## RESULTS

### Inhibitory effects of *R. angustifolia* extracts against *P. gingivalis*

Based on the well diffusion test presented in Fig. 1, all *R. angustifolia* extracts effectively suppressed the growth of *P. gingivalis* in a concentration-dependent manner. Inhibition zone measurements indicated that the highest antimicrobial activity occurred at a concentration of 100 mg/mL for the chloroform fractionated extract of *R. angustifolia*, as shown in Table 1. Among the three extracts tested, the chloroform fractionated extract exhibited the largest inhibition zone, followed by the crude methanol extract (9.30 ± 0.26 mm) and the hexane fractionated extract (8.19 ± 0.27 mm). These results demonstrate that the chloroform fractionated extract of *R. angustifolia* most effectively inhibited the growth of *P. gingivalis*, while the negative control (1% DMSO) showed no inhibition.

In comparison, the inhibition zone for 0.12% CHX was 13.34 ± 0.73 mm, while the chloroform fractionated extract at 100 mg/mL demonstrated an inhibition zone of 11.65 ± 0.62 mm. Thus, these findings suggest that its efficacy is comparable to that of 0.12% CHX, a widely used antibacterial agent for oral diseases. Although CHX exhibited a slightly larger zone, the difference is not substantial, indicating that *R. angustifolia* may serve as an effective alternative or adjunct in managing oral infections.

### Determination of MIC and MBC

Given that *P. gingivalis* is a significant pathogen involved in the development and pathophysiology of periodontal infections, the primary aim of this study was to further assess the antimicrobial activity of the extracts against this oral bacterium. Based on the initial antimicrobial screening, the chloroform fractionated extract of *R. angustifolia* demonstrated the most potent inhibitory effect against *P. gingivalis*. Therefore, this extract was selected for all subsequent analyses in the study.

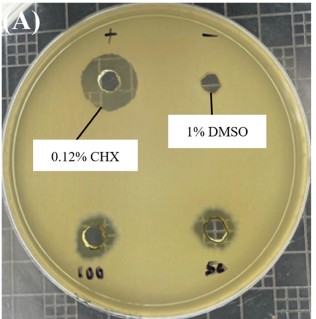
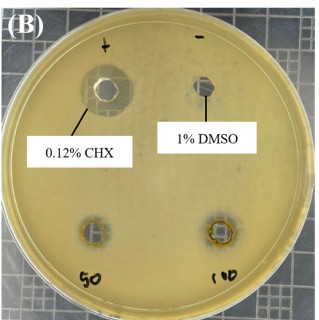
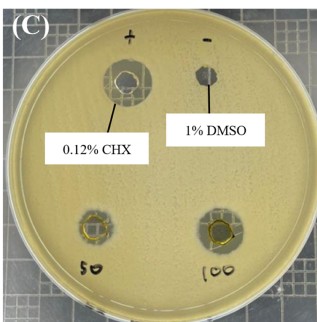

**Figure 1 Antimicrobial effect of *R. angustifolia* extracts against *P. gingivalis*.** Bacterial cells were treated with 50 and 100 mg/mL of (A) crude methanol extract, (B) hexane and (C) chloroform fractionated extract, respectively. The representative images display the measured inhibition zone (in mm) of the extracts; 0.12% CHX (used as a positive control); and 1% DMSO (served as the negative control). Larger zones indicate greater inhibition.                

**Table 1 The inhibition zone diameters of *R. angustifolia* extracts against *P. gingivalis*.**

| Microorganism | | Inhibition zone diameter (mm) | | |
|---|---|---|---|---|
| | | *R. angustifolia* extracts | | |
| | | Crude methanol | Hexane | Chloroform |
| *P. gingivalis* ATCC 33277 | 50 mg/mL | 8.12 ± 0.77* | 6.98 ± 0.26* | 8.32 ± 1.08* |
| | 100 mg/mL | 9.30 ± 0.26 | 8.19 ± 0.27* | 11.65 ± 0.62 |
| | 0.12% CHX[a] | 13.79 ± 0.71 | 13.45 ± 0.79 | 13.34 ± 0.73 |

**Notes:**
[a] 0.12% CHX served as the positive control, whilst 1% DMSO as the negative control against *P. gingivalis*. Experiments were conducted in triplicate and tabulated values are represented as mean ± standard deviation. Statistically significant with Kruskal-Wallis ($p$-value < 0.05) and Dunn's *post-hoc* analysis (*$p < 0.05$ *vs.* control according to analysis of variance).

**Table 2 MIC & MBC of chloroform fractionated extract of *R. angustifolia* against *P. gingivalis*.**

| Microorganism | Chloroform fractionated extract of *R. angustifolia* (mg/mL) | |
|---|---|---|
| | MIC | MBC |
| *P. gingivalis* | 6.25 | 6.25 |

**Note:**
MIC, minimum inhibitory concentration; MBC, minimum bactericidal concentration.

The MIC and MBC values of the chloroform fractionated extract of *R. angustifolia* (Table 2) were determined using serial microdilution, with concentrations ranging from 0.39 to 50 mg/mL. The antimicrobial activity of the extract was compared against untreated *P. gingivalis* bacterial cells in 1% DMSO (which served as the negative control). The MIC was found to be at 6.25 mg/mL, as this concentration resulted in a significant reduction in *P. gingivalis* growth compared to the untreated sample. The negative control exhibited no inhibition, confirming its suitability as a baseline comparison. Further analysis revealed that the MBC of the chloroform fractionated extract of *R. angustifolia* against *P. gingivalis* was determined to be 6.25 mg/mL, as no bacterial growth was observed at this level.

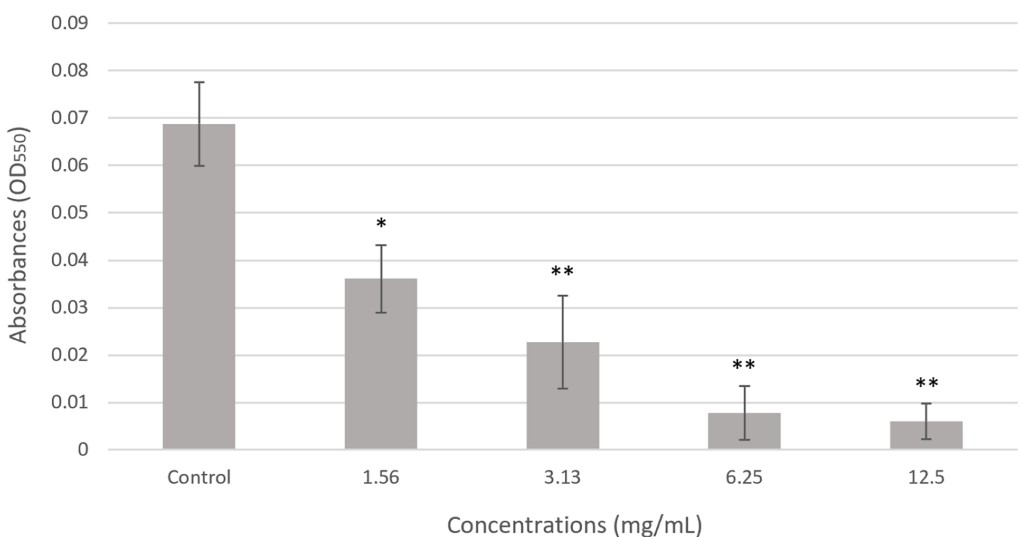

**Figure 2 Effect of chloroform fractionated extract of *R. angustifolia* against *P. gingivalis* biofilm *via* the crystal violet assay.** The untreated *P. gingivalis* biofilm, is the negative control. Experiments were conducted in triplicate and values are represented as mean ± standard deviation. Statistically significant with ANOVA ($p$-value < 0.05) and Tukey's HSD *post hoc* analysis (*$p$ < 0.05, **$p$ < 0.001 *vs.* control according to analysis of variance).

## Effect of chloroform fractionated extract of *R. angustifolia* against *P. gingivalis* biofilm biomass

As shown in Fig. 2, the overall biomass of *P. gingivalis* biofilms treated with various concentrations of chloroform fractionated extract of *R. angustifolia* was performed using crystal violet staining. The extract demonstrated notable antibiofilm effects by suppressing the formation of biofilm in *P. gingivalis*. A clear, concentration-dependent decrease in absorbance was observed, indicating the efficacy of the extract in reducing the biofilm biomass. Notably, the biofilm biomass of *P. gingivalis* was remarkably reduced by up to 91% at the highest concentration of chloroform fractionated extract of *R. angustifolia*. This result was consistent with the MIC determined in Table 2, confirming that the chloroform fractionated extract of *R. angustifolia* effectively disrupted the formation of *P. gingivalis* biofilm compared to the control.

## Live/dead cell viability assay using fluorescence microscope

LIVE/DEAD® BacLight™ viability staining was conducted to evaluate the impact of the chloroform fractionated extract of *R. angustifolia* on *P. gingivalis* biofilm (Fig. 3). In the absence of the extract, the untreated *P. gingivalis* biofilm displayed a dense and highly uniform structure after 24 h of incubation. In contrast, increasing concentrations of the extract led to marked sparsity and dispersion of the biofilm, indicating a significant reduction in viable cells. As a result, biofilm formation was substantially inhibited, with a notable decrease in bacterial presence at higher concentrations of the chloroform fractionated extract of *R. angustifolia*.

Similar results were observed at 48 h, where the untreated *P. gingivalis* biofilm remained dense and uniform, while higher extract concentrations led to increased biofilm disruption

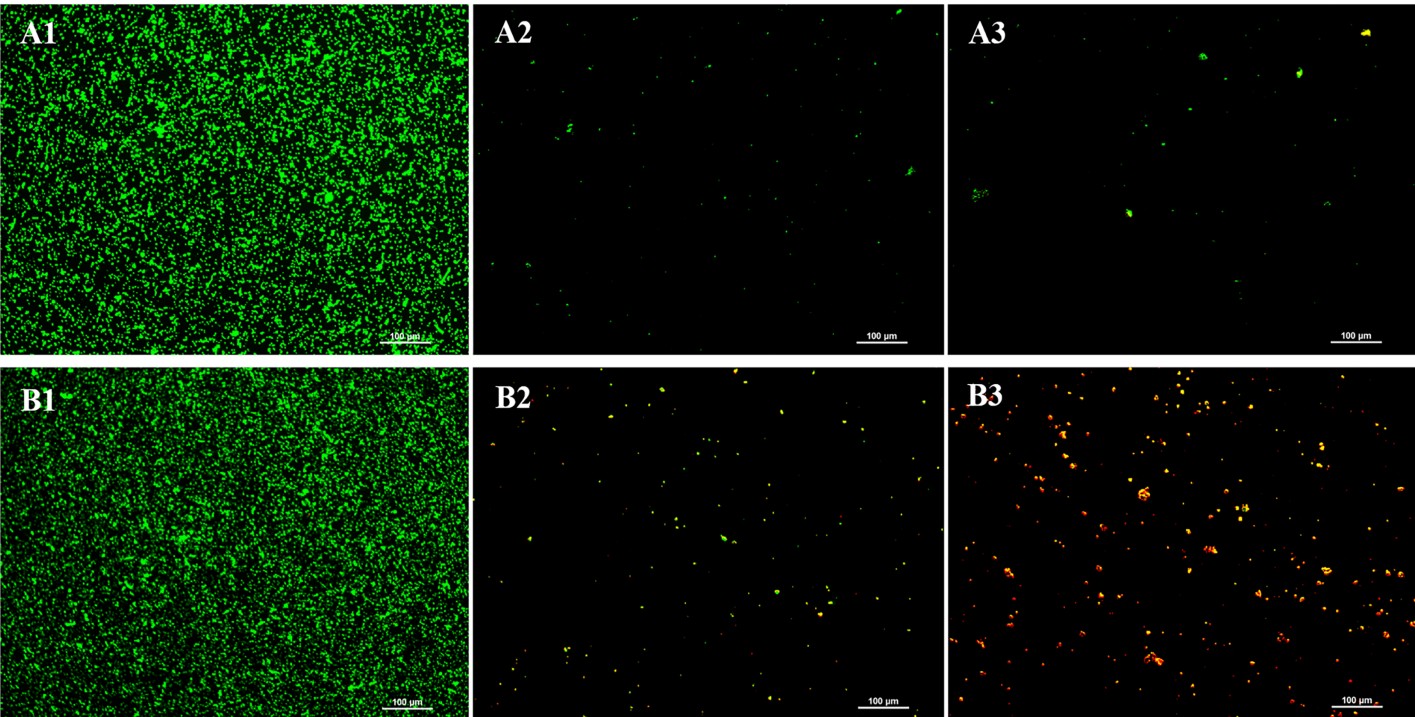

**Figure 3 Representative images from LIVE/DEAD® BacLight™ viability staining of untreated and treated *P. gingivalis* with chloroform fractionated extract of *R. angustifolia*.** Biofilm images at (A) 24 h and (B) 48 h incubation time. *P. gingivalis* biofilm (1) without any treatment, *P. gingivalis* biofilm after treated with (2) 3.13 mg/mL and (3) 6.25 mg/mL of chloroform fractionated extract of *R. angustifolia*. Bacteria that are viable/alive cells were stained green and non-viable/dead cells were stained red. (Magnification: 10x, scale bar: 100 μm).

and cell death. Significantly, an abundance of lysed cells and cellular debris was visible at the higher concentration of 6.25 mg/mL. These findings demonstrate that the chloroform fractionated extract of *R. angustifolia* is capable of disrupting *P. gingivalis* biofilm formation and viability in a concentration-dependent manner.

## The effects of the *R. angustifolia* extract on *P. gingivalis* cell morphology *via* FE-SEM

FE-SEM imaging (Fig. 4) was conducted to examine morphological alterations in *P. gingivalis* exposed to the chloroform fractionated extract of *R. angustifolia*. Untreated *P. gingivalis* exhibited a characteristic of smooth cell surface that appeared as tiny grape-like clusters with intact cell membranes. In contrast, the *P. gingivalis* biofilm treated with 6.25 mg/mL of the chloroform fractionated extract of *R. angustifolia* displayed significant changes in cellular morphology. The treated cells displayed an unravelled, crater-like appearance of the cell walls, as along with swollen, elongated, and lysed forms that aggregated into large clusters of cellular debris. Therefore, the FE-SEM analysis conclusively demonstrated that the chloroform fractionated extract of *R. angustifolia* completely inhibited *P. gingivalis* cells. At this MIC concentration, the integrity of the *P. gingivalis* bacterial cell membrane was compromised, resulting in extensive cell lysis and disruption of cellular morphology.

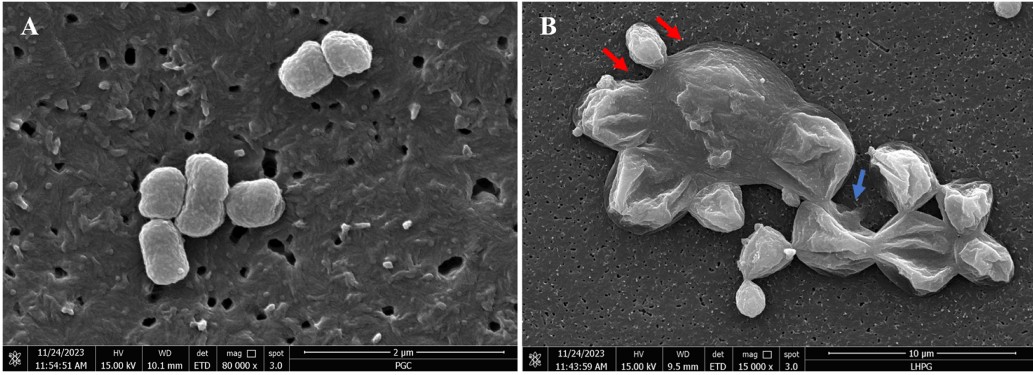

**Figure 4** FE-SEM images of (A) non-treated and (B) treated *P. gingivalis* with 6.25 mg/mL of chloroform fractionated extract of *R. angustifolia*. Red arrows: Swollen rods, Blue arrow: Cells ruptured to form deep craters. Magnification: 80kx (left), 15kx (right).

## DISCUSSION

The present study reveals that the crude methanol extract, hexane, and chloroform fractionated extracts of *R. angustifolia* exhibit inhibitory activity against *P. gingivalis*. Among these, the chloroform fractionated extract of *R. angustifolia* was found to be the most effective in inhibiting the growth of *P. gingivalis*, leading to its selection for further investigation. This report is the first to document the antimicrobial susceptibility of *R. angustifolia* extracts against *P. gingivalis*. The antimicrobial effect of the extracts was dose-dependent, with higher concentrations of the chloroform fractionated extract of *R. angustifolia* displaying greater bactericidal activity. A previous report identified twelve components isolated from the chloroform fractionated extract of *R. angustifolia* (*Richardson, Aminudin & Abd Malek, 2017*). These compounds were graveoline, kokusaginine, bergapten, moskachan B, moskachan D, chalepensin, rutamarin, arborinine, chalepin and neophytadiene. Most of these compounds belong to the classes of alkaloids, furanocoumarins, and dihydrofuranocoumarins, which possess significant antiviral, anti-inflammatory, antioxidant, and antiproliferative properties (*Bailly, 2023*). Specifically, rutamarin and chalepin are categorised as furanocoumarins (*Richardson et al., 2016*). Current research indicates that coumarin derivatives, recognised for their antioxidant, antifungal, and antibacterial properties, hold promise as therapeutic agents across various medical applications. Structural modifications, such as the incorporation of nitrogen atoms and azomethane groups, are thought to enhance antibacterial activity by increasing membrane permeability, thereby contributing to potent bactericidal effects (*Wahyuni et al., 2023*). These compounds likely benefit from increased lipophilicity, which enhances their ability to penetrate bacterial membranes. This lipophilic characteristic may also interfere with bacterial metabolic processes by inactivating essential enzymes and denaturing cellular proteins, ultimately leading to bacterial cell death (*Betti et al., 2024*). Additionally, recent studies have revealed that coumarin derivatives exhibit significant cytotoxicity towards bacterial DNA, potentially due to their modification of bacterial membrane components, including lipopolysaccharides (LPS). This modification may

activate enzymes such as topoisomerases and helicases, destabilizing the DNA structure and obstructing replication, which culminates in bacterial cell apoptosis (*Koszelewski et al., 2023*).

Furthermore, graveoline, one of the isolated components identified from the chloroform fractionated extract of *R. angustifolia* (*Richardson et al., 2016*), exhibits antibacterial, anticancer, antifungal, and spasmolytic effects. Graveoline is a quinolone alkaloid that interacts with bacterial cell membranes, affecting their function and causing cell lysis. Alkaloids can suppress bacterial growth by interfering with nucleic acid, protein synthesis, and metabolism. Additionally, these compounds may disrupt essential metabolic processes and energy sources for various enzymatic reactions in bacteria, such as adenosine triphosphate (ATP), thereby inhibiting growth. Alkaloids can also promote the generation of reactive oxygen species (ROS), which damage bacterial cells (*Yan et al., 2021*). Hence, *R. angustifolia* harbours numerous bioactive compounds, showcasing significant potential for therapeutic applications in various diseases, including periodontal diseases.

The periodontal pathogen *P. gingivalis* is known as one of the late colonisers of the oral cavity, which possess a range of virulent characteristics and robust biofilm-forming capabilities that facilitate persistent infection at the gingival site (*Xu et al., 2020*). These late colonisers consist of slower-growing, obligate anaerobes, and Gram-negative species, such as *P. gingivalis*, along with other members of the red complex, *T. forsythia* and *T. denticola*, which are typically found at elevated levels in diseased conditions (*Siddiqui et al., 2022*).

Gram-positive and Gram-negative bacteria are primarily distinguished by the thickness of their cell walls and the presence of an outer membrane, which is unique to Gram-negative bacteria. The thickness of the cell walls of Gram-positive and Gram-negative bacteria ranges from 20 to 80 nm and 1.5 to 10 nm, respectively (*Bismelah et al., 2022*). Despite having thicker cell walls, Gram-positive bacteria can absorb foreign chemicals, such as bioactive compounds, making them more susceptible to elimination. In contrast, the effectiveness of some antibiotic treatments against Gram-negative bacteria is impaired by the presence of an outer membrane composed of lipopolysaccharides and phospholipids (*Fivenson et al., 2023*). This additional membrane layer functions as an effective permeable barrier, preventing antibiotic molecules from entering the bacterial cell.

The presence of virulence factors in various periodontopathic bacteria poses significant challenges in treating infections. For example, *P. gingivalis* and *T. forsythia* are characterised by high proteolytic and peptidolytic activities, which can hydrolyse antibiotics, leading to the inactivation of their antimicrobial properties (*Bismelah et al., 2022*). These factors may explain why higher concentrations of plant extracts are required to effectively inhibit *P. gingivalis* in susceptibility tests. Therefore, the extract of *R. angustifolia* could serve as an adjunctive treatment for periodontitis, as it effectively targets the growth of late colonisers, thereby helping to prevent the progression of periodontal infections.

Correspondingly, the chloroform fractionated extract of *R. angustifolia* showed promising results for MIC and MBC, completely inhibiting *P. gingivalis* at a concentration

of 6.25 mg/mL. Furthermore, the biofilm assay provided evidence that the chloroform fractionated extract of *R. angustifolia* exhibits inhibitory effects on the formation of *P. gingivalis* biofilms. This significant discovery suggests that the extract has potential as a viable method for mitigating the metabolic activity of *P. gingivalis* within biofilms. Biofilms are complex three-dimensional structures composed of microorganisms bound together by an extracellular matrix (*Mitchell, 2011*). Once microorganisms form biofilms, they become increasingly challenging to control and eradicate. According to *Raut & Karuppayil (2016)*, biofilms are associated with over 80% of microbial illnesses in humans. Since periodontal disease involves the formation of biofilms on gingival tissues, disrupting the biofilm formed by *P. gingivalis* is critically important (*Bustamante et al., 2020*). The ability of the chloroform fractionated extract of *R. angustifolia* to interfere with biofilm formation at varying concentrations is crucial for preventing *P. gingivalis* from recolonising the periodontal pocket, as this treatment eliminates the protective layer of the subgingival biofilm (*Bostanci & Belibasakis, 2012*).

Moreover, the outcomes of our investigation closely correlate with a recent study that have demonstrated the ability of *R. angustifolia* extracts in diminishing the formation of biofilm by *S. mutans* at concentrations of 20%, 50%, and 100% (*Noer, Abinawanto & Bowolaksono, 2021*). In general, the antimicrobial activities of plants are attributed to their secondary metabolites. Previous studies has confirmed that *R. angustifolia* extract contains steroids, flavonoids, tannins, and quinones, suggesting its potential as an antibiofilm agent (*Lu et al., 2019*). Compounds such as flavonoids in *R. angustifolia* extract have been extensively studied for their bioactivity in inhibiting both Gram-positive and Gram-negative bacteria through the inactivation of extracellular proteins (*Bismelah et al., 2022*). Recent studies indicate that the antibacterial activities of plant flavonoids against Gram-negative bacteria are related to their lipophilicity. The cell envelope of Gram-negative bacteria contains an inner lipophilic membrane and an outer hydrophilic membrane, while Gram-positive bacteria possess only a cell membrane. Consequently, it is suggested that plant flavonoids employ multiple mechanisms against Gram-negative bacteria, with the cell membrane serving as a critical site of action, in contrast to Gram-positive bacteria, which are primarily affected at the cell membrane. Furthermore, DNA gyrase is another important target of plant flavonoids against Gram-negative bacteria. Thus, plant flavonoids can damage the cell membrane and inhibit DNA gyrase, representing a dual mechanism of action against Gram-negative bacteria (*Yan et al., 2024*).

Live/dead cell viability assays demonstrated that the chloroform fractionated extract of *R. angustifolia*, at both sub-MIC and MIC levels, induced a significant reduction in the viability of *P. gingivalis* compared to the control group. The biofilm formation of *P. gingivalis* was markedly inhibited by the chloroform fractionated extract of *R. angustifolia*, which led to a greater dispersion of biofilm structures and an increase in cell death, particularly at a concentration of 6.25 mg/mL after 48 h of incubation. Moreover, non-viable/dead (red fluorescence) cells were evidently in greater proportion than viable/live (green fluorescence) cells. These findings suggest that the chloroform fractionated extract of *R. angustifolia* exhibits biocidal effects on the *P. gingivalis* biofilm, as evidenced by the visualization of dead microorganisms. Similar results were reported from

a previous study by *He et al. (2020)*, which investigated the impact of quercetin on *P. gingivalis* aggregation and biofilm formation.

FE-SEM images revealed morphological changes in the cell structure of *P. gingivalis* after treated with the chloroform fractionated extract of *R. angustifolia*. The results indicated that the extract caused damage to the cell membrane, leading to subsequent cell death. Active bioactive compounds, such as graveolinine, flavonoid, chalepin, and rutamarin, have been identified in *R. angustifolia* extracts and are known to contribute to alterations in cell morphology in cancer cells (*Kamal et al., 2021*; *Richardson, Aminudin & Abd Malek, 2017*). Rutamarin and chalepin were reported to exhibit similar mechanisms, resulting in cell shrinkage, the formation of apoptotic bodies, and rounding of cancer cells (*Richardson et al., 2016*; *Suhaimi, Hong & Abdul Malek, 2017*). Furthermore, flavonoids have been shown to damage the cytoplasmic membrane, inhibit nucleic acid production, and disrupt energy metabolism of bacterial cells (*Shamsudin et al., 2022*). These factors may collectively contribute to the cell death of *P. gingivalis*. In contrast, untreated control cells grew exponentially while maintaining an intact structure. *He et al. (2020)* observed similar effects, where quercetin-treated *P. gingivalis* exhibited cell elongation from its coccobacillus shape, followed by cell shrinkage and eventual lysis due to deformed cell walls. Notably, the mechanism of action of the extract resembles that of chlorhexidine, in which low concentrations of chlorhexidine disrupt the integrity of the cell wall, ultimately resulting in damage. Consequently, chlorhexidine penetrates the cell, inducing leakage of the cytoplasm that leads to cell death (*He et al., 2020*).

The initial findings on the antimicrobial activity of the *R. angustifolia* extracts against the monospecies bacterium *P. gingivalis* are promising, indicating potential effectiveness in inhibiting this essential periodontal pathogen. However, since *P. gingivalis* is only one species within the complex oral microbiome, these results may not fully reflect the impact of the extracts on other oral microorganisms that contribute to oral health and disease. Therefore, to enhance the robustness and generalizability of the findings, future studies should encompass a broader range of oral bacterial species, such as *T. denticola, T. forsythia, S. mutans*, and *Fusobacterium nucleatum*. This approach will provide a better understanding of the antimicrobial potential across the diverse microbial community in the oral cavity and enhance the relevance of the findings.

In this present study, 0.12% CHX was used as the positive control. CHX is a widely used antibacterial agent in dentistry, known for its broad-spectrum antibacterial properties against various pathogens, including periodontal bacteria such as *P. gingivalis*. The concentration of 0.12% CHX is frequently incorporated into mouthwashes and other oral care products, establishing it as a standard concentration commonly used in numerous studies as a positive control in antibacterial tests. Utilizing 0.12% CHX as a positive control provides researchers with a reliable benchmark to compare the antibacterial effectiveness of natural products or extracts against a well-established and clinically validated antibacterial agent. This approach ensures that the findings are relevant to clinical practice.

Furthermore, while the *in vitro* results are encouraging, it is essential to thoroughly evaluate the biocompatibility of the antimicrobial agents to ensure their safety for human use. This evaluation should include testing for potential cytotoxicity and assessing the

effects on normal human oral cells such as gingival and periodontal cells. Additionally, clinical efficacy should be explored through further studies utilizing human cellular models to replicate real-life conditions. Moreover, in addition to *in vitro* studies, *in vivo* research using animal models, such as rodents, is crucial for evaluating the effectiveness of the extracts in a living organism. This evaluation should consider factors such as bioavailability, immune response, and long-term safety. Ultimately, these additional studies will offer a more comprehensive understanding of the therapeutic potential of the extracts while ensuring their safety and effectiveness for future application in human oral healthcare.

## CONCLUSIONS

In conclusion, this *in vitro* study provides new evidence that *R. angustifolia* exhibits antimicrobial activity against periodontal pathogens, notably *P. gingivalis*. These findings suggest that *R. angustifolia* has the potential to be used as an alternative treatment for periodontal disease and warrants further investigation.

## ACKNOWLEDGEMENTS

The authors would like to express their appreciation to Professor Dr. Ian Charles Paterson, a native English speaker, for his careful proofreading and feedback on the language and grammar of this manuscript. They also extend their gratitude to Mr. Anuar and the laboratory staff of Balai Ungku Aziz Research Laboratory (BUARL), Faculty of Dentistry, Universiti Malaya, for their assistance during the course of this study.

### Funding

This research was supported by the Ministry of Higher Education Malaysia under the Fundamental Research Grant Scheme (FRGS/1/2020/SKK0/UM/02/14). The funders had no role in study design, data collection and analysis, decision to publish, or preparation of the manuscript.

### Grant Disclosures

The following grant information was disclosed by the authors:
Ministry of Higher Education Malaysia: FRGS/1/2020/SKK0/UM/02/14.

### Competing Interests

The authors declare that they have no competing interests.

### Author Contributions

- Husna Hazirah Bakri conceived and designed the experiments, performed the experiments, analyzed the data, prepared figures and/or tables, authored or reviewed drafts of the article, and approved the final draft.
- Syarifah Nur Syed Abdul Rahman conceived and designed the experiments, analyzed the data, authored or reviewed drafts of the article, and approved the final draft.

- Zarith Safinaz Dol Bakri analyzed the data, authored or reviewed drafts of the article, and approved the final draft.
- Elly Munadziroh analyzed the data, authored or reviewed drafts of the article, and approved the final draft.
- Wan Himratul Aznita Wan Harun conceived and designed the experiments, analyzed the data, authored or reviewed drafts of the article, and approved the final draft.

## Data Availability

The raw datasets and images are available in the Supplemental Files.

## Supplemental Information

Supplemental information for this article can be found online at http://dx.doi.org/10.7717/peerj.18751#supplemental-information.

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
