# Peer review of "Antimicrobial activity of Ruta angustifolia L. Pers against periodontal pathogen: Porphyromonas gingivalis"

_PeerJ, doi:10.7717/peerj.18751_

## Round 0.1 · original submission · Major Revisions

· Academic Editor

Major Revisions

Dear authors,
The current work is well-structured, with a clear research plan addressing important questions. However, a few revisions and clarifications are necessary. First, the manuscript could improve in explaining the use of spectrophotometer readings for MIC and MBC determinations, as MICs should ideally be determined visually according to EUCAST and CLSI guidelines. Also, an experiment replicating organic extraction without the plant is suggested to ensure the differences between solvents are accurately assessed. Further clarification on measuring precise values like 11.83 mm is needed. Overall, the work is promising, but some additional experiments and revisions would strengthen it.

·

Basic reporting

Current work is interesting since there is a good experiments planifications in order to give answers to their research question.
In general terms work is well--conducted, clear and references are completely associated.

Experimental design

About experiments, as I say above, is well conducted just some points.
1. MIC and MBC are the same, this is because readers of test were performed with spectrophotometer, MICs recommendations accordin EUCAST and CLSI must be done a simpke sight, actually that is part of definion, the the use of an accurate system as spectrophotometer will be most accurate, then the possibility of same MIC and MBC is high.
2. I´d like to ask for an experiment where authors can to emulate organic extraction (without the plant) just to have a real parameter about the differences between an organic solvent and another is related with type and quantitative of products and not as a residual activity.
Is there any way to demostrate, in general and visual termns, these concentrations differences?

Validity of the findings

In general terms work has been well conducted I just want to have more information thats why I ask for a couple of experiments to improve the evidence.
And one doubt about. How do you meassure 11.83 mm, I mean how do you measure those centecimals. Even if you use a Vernier the expresion of centesimals...

Additional comments

There is no problem to be accepted after some experiments

Reviewer 2 ·

Basic reporting

Comments:
This is interesting on the potential use of the Ruta angustifolia Pers for oral bacterial infections.
The manuscript needs some revision. Comments are as follows.

Abstract:
The last line is Conclusion rather than Discussion.


Introduction:

Line 71-75 need some revision. It is better to add more on the use of Herbal products in the oral health improvement and alveolar bone protection.

Experimental design

Method:
The authors used the minimum inhibitory concentration (MIC) & minimum bactericidal concentration (MBC) to test the anti-bacterial properties. How many times the experiments are repeated to see the reliability and validity?

As a control, they used 0.12% CHX was used as the positive control. Add rationale behind this.

Please add details on the sample size to support the statistical test used.

Validity of the findings

The authors mention that R. angustifolia extracts efficiently suppressed the growth of P. gingivalis in a concentration-dependent manner. Did the Ruta angustifolia Pers show better results than the CHX? Please discuss more on this.

Additional comments

Discussion:
Discuss the mechanism of antibacterial actions.

Please add limitations of this research.
Add future recommendations.

---

## Round 0.2 · Minor Revisions

· Academic Editor

Minor Revisions

Dear authors

Further considerations were raised by the reviewers to improve your manuscript. Please try to address all these considerations to avoid a further round of revision.

Thanks

·

Basic reporting

Authors have attendeds the most of comments made previously.
Just a little things about stylie.
Line 111, 113, 125, 137, 141, 142, 143, 156, 158, 160, 162, 167, 176, 188, 216, 219, 264, 272, 290, . To remove double 0 after point

Experimental design

Mentioned before, well designed, well conducted

Validity of the findings

Results according methods.

Additional comments

Not problem to be accepted

·

Basic reporting

The manuscript generally meets the standards, with a clear research aim and overall good clarity of language. However, there are areas where improvements can enhance the manuscript:

1. Introduction: The aim of the study is clearly stated; however, the hypothesis is not explicitly mentioned and should be articulated to provide a clearer focus. Additionally, the introduction could be strengthened by expanding on the challenges of treating periodontal disease, the critical role of P. gingivalis in the disease, and the importance of finding new therapeutic alternatives. While the authors have provided context, further discussion of chlorhexidine’s limitations, would add value.

2. References: There is a need for additional and more recent references, particularly in lines 71–79, where the references are somewhat outdated (e.g., from 2003). If possible, update these and other citations with newer research that is relevant to the topic.

3. Grammar and clarity: Minor grammatical improvements are suggested:
- Line 69: Change "Thus, finding safe and effective therapeutic agent for oral health is deemed necessary" to "Thus, finding a safe and effective therapeutic agent for oral health is deemed necessary."
- Line 131: "gram stained" should be corrected to "Gram-stained."
- Line 168 and 227: Specify whether wells were incubated under anaerobic or aerobic conditions to clarify methods.
- Line 199: Specify the number of washes performed on the wells.
- Line 294-298: The description of the minimum bactericidal concentration (MBC) test results would benefit from a clearer and more precise rewrite. I suggest the authors revise the statement: “Further analysis of the MBC test showed bacterial growth at 3.13 mg/mL, indicating that MBC was not detected at this concentration. Therefore, the MBC of the chloroform fractionated extract of R. angustifolia against P. gingivalis was determined to be 6.25 mg/mL, as no bacterial growth was observed at this level.” for improved clarity and flow. For example, it could be rephrased to better convey the findings and their significance.

4. Methodological clarity: while generally adequate, methodology section could be more detailed for better replication. Specifically, the authors should clarify the origin of the R. angustifolia plants used in the study. Were the plants purchased? If so, from where? Providing this information would ensure transparency and reproducibility in future studies. Moreover, the section on the “Live/Dead cell viability assay using fluorescence microscopy” (lines 207–223) lacks clarity regarding the type of plate used and the microscope objective/lens and fluorescence settings. This information should be provided to improve replicability.

5. Figures and tables:
- In Table 1, using “oral microorganism” may give the impression of clinical strains being used. Instead, use “microorganisms” or “bacteria,” and specify P. gingivalis with its ATCC number.

6. Discussion: The first paragraph lacks citations, which should be added to strengthen the claims. Additionally, consider analyzing the extract components of R. angustifolia, as the abundance of compounds like graveoline can vary based on extraction methods, plant origin, etc.

Overall, the manuscript is well-organized and clear, with good relevance to the field. Addressing these points would enhance clarity, comprehensiveness, and reporting standards.

Experimental design

The experimental design appears generally appropriate for assessing the antimicrobial activity of R. angustifolia extracts against P. gingivalis. However, there are areas that could benefit from additional clarity and detail. In particular, the methods section would be strengthened by specifying the origin of the R. angustifolia plants used in the study. Providing such information ensures transparency and reproducibility.

Additionally, the methodology around biofilm assays and culture conditions could be more detailed. For example, it would be helpful to specify whether the wells were incubated under anaerobic or aerobic conditions (lines 168 and 227). Moreover, clarifying whether fresh culture media were renewed during the 48-hour incubation period for biofilm viability testing (line 190) - and if not, please explain why - would improve the understanding of experimental consistency.

Further clarity on certain aspects, such as the exact plate type used during fluorescence microscopy and field emission scanning electron microscopy (FE-SEM), would also aid in replicating the experiments.

Validity of the findings

The findings are generally valid and the underlying data are robust. Moreover, the conclusions are well-stated and appropriately linked to the original research question.

---

## Round 0.3 · accepted · Accept

· Academic Editor

Accept

Dear authors,

Congratulations on your work. The manuscript has been revised thoroughly, addressing all recommendations made by the reviewers. The basic reporting is clear, and all suggestions have been implemented effectively. The experimental design aligns well with the objectives, and the validity of the findings is robust and well-demonstrated through appropriate methodologies.

·

Basic reporting

Authors attended all recommendations made

Experimental design

According design

Validity of the findings

According methods, well demostrated

Additional comments

I´d like to congratulate author to attend all recomendations, but also to reviewer 2 for amazing work beyond the form, with impact in experiments and grammar.
I have learn from the reviewer also. Thank you so much.